# Clinical Outcomes in COVID-19 Patients Treated with Immunotherapy

**DOI:** 10.3390/cancers14235954

**Published:** 2022-12-01

**Authors:** Haris Hatic, Kristine R. Hearld, Devika Das, Jessy Deshane

**Affiliations:** 1Suburban Hematology Oncology Associates, 631 Professional Drive Suite 450, Lawrenceville, GA 30046, USA; 2Department of Health Services Administration, University of Alabama at Birmingham, Birmingham, AL 35294, USA; 3Department of Hematology-Oncology, University of Alabama, 2000 6th Ave S, Birmingham, AL 35233, USA; 4Section Chief for Oncology, Birmingham VA Medical Center, Birmingham, AL 35233, USA; 5Division of Pulmonary Allergy and Critical Care Medicine, Department of Medicine, University of Alabama, Birmingham, AL 35233, USA

**Keywords:** COVID-19, immunotherapy, cancer, immune checkpoint inhibitors

## Abstract

**Simple Summary:**

Patients with cancer who contract COVID-19 are very vulnerable to increased complications and illness while actively being treated with chemotherapy or immune checkpoint inhibitors (ICIs). The aim of this retrospective review was to describe the disease course and identify specific risk factors and overall outcomes in COVID-19-affected patients who are also diagnosed with cancer. We examined whether treatment (chemotherapy vs. ICIs) was associated with clinical outcomes, including hospitalization rates, ICU admissions, and any-cause mortality. A total of 121 patients were examined in this study, and 61 (50.4%) received immunotherapy treatment within 12 months. COVID-19-related ICI mortality was higher compared to patients receiving chemotherapy, but patients with better functional status and COVID-19 vaccination had reduced mortality. ICI cessation or delay is unwarranted as long there has been a risk–benefit assessment undertaken with the patient. However, further investigation still needs to be undertaken with a larger cohort, with an emphasis on timing and outcomes between ICI therapy and COVID-19 infection.

**Abstract:**

Introduction: The full impact of COVID-19 infections on patients with cancer who are actively being treated with chemotherapy or immune checkpoint inhibitors (ICIs) has not been fully defined. Our goal was to track clinical outcomes in this specific patient population. Methods: We performed a retrospective chart review of 121 patients (age > 18 years) at the University of Alabama at Birmingham from January 2020 to December 2021 with an advanced solid malignancy that were eligible to be treated with ICIs or on current therapy within 12 months of their COVID-19 diagnosis. Results: A total of 121 patients were examined in this study, and 61 (50.4%) received immunotherapy treatment within 12 months. One quarter of the patients on ICIs passed away, compared to 13% of the post-chemotherapy cohort. Patients who were vaccinated for COVID-19 had lower mortality compared to unvaccinated patients (*X*^2^ = 15.19, *p* < 0.001), and patients with lower ECOG (0.98) were associated with lower mortality compared to patients with worse functional status (0.98 vs. 1.52; t = 3.20; *p* < 0.01). Conclusions: COVID-19-related ICI mortality was higher compared to patients receiving chemotherapy. However, ICI cessation or delay is unwarranted as long there has been a risk–benefit assessment undertaken with the patient.

## 1. Introduction

In December 2019, coronavirus disease 2019 (COVID-19), caused by severe acute respiratory syndrome coronavirus 2 (SARS-CoV-2), emerged in China and subsequently led to a global pandemic in early 2020 [1]. As of November 2022, the COVID-19 pandemic has caused over 1,000,000 deaths and almost 98 million cases in the United States alone [2]. As a result, it has created an immense strain on the overall healthcare system. 

To alleviate further stress on the healthcare system, it has become essential to identify subsets of vulnerable populations that are at an increased risk of severe complications and illness from COVID-19. At this time, we know that older age, immunodeficiency and increased comorbidities (hypertension, diabetes, coronary artery disease, chronic obstructive pulmonary disease) can increase mortality from COVID-19 [3,4]. Moreover, it has been previously suggested that individuals receiving systemic anticancer treatment, such as chemotherapy, have an increased risk of COVID-19-related mortality due to a weakened immune system [5,6]. However, further studies are required to differentiate treatment effects from a patient’s general disease course on the clinical outcomes of COVID-19 [7].

Immunotherapy (IMT) is a type of cancer treatment that enhances the body’s own defense mechanism to combat malignant cells by releasing the “brakes” on the immune system [8]. Immune checkpoint inhibitors (ICIs) are immunomodulatory antibodies that result in T-cell activation through a complex cascade of activating pathways that lead to cancer cell death [9]. It begins with antigen presentation on the surfaces of antigen-presenting cells (APCs) via T-cell receptors (TCR) and major histocompatibility complex (MHC) [10,11]. Subsequently, there is the co-stimulation of immune checkpoints, which provides suppression of the immune response [12]. The balance between stimulatory and inhibitory signals is driven by ligands such as cytotoxic T-lymphocyte-associated protein-4 (CTLA-4) and programmed cell death-1 (PD-1) [13]. CTLA-4 reduces signaling via its co-stimulatory receptor, CD28, by increasing the activation threshold of T-cells [11,13]. This reduces responses to self- and tumor antigens [13]. PD-1 binds to programmed death-ligand 1 (PD-L1), which results in the maintenance of anti-tumor T-cell effector function, T-cell inhibition and cell proliferation (Figure 1) [13]. ICIs include anti-CTLA-4 (ipilimumab), anti-PD-1 (nivolumab, pembrolizumab, cemiplimab or dostarlimab) and anti-PD-L1 (atezolizumab, avelumab and durvalumab) inhibitors [11]. PD-L1/PD-1 blockade therapy significantly improves the durable response rate and prolongs long-term survival, with limited adverse effects in both monotherapy and combination therapy [14,15,16,17,18,19,20,21].

As already noted, ICIs have changed the treatment landscape of both hematological and solid malignancies [11,22,23,24]. Tumors with T-cell infiltration have increased recognition by the immune system, which leads to an improved ICI response. These malignancies are termed “hot” tumors and include the bladder, kidney, head and neck, melanoma and non-small cell lung carcinoma (NSCLC) [25,26,27]. However, tumors with fewer infiltrating T-cells, which reduces the response to ICIs, are termed “cold” and comprise breast, prostate, ovarian and pancreatic cancer [28,29,30]. Current research is focused on exploring potential mechanisms that can turn “cold” tumors into “hot” [31].

The recent SARS-CoV-2 pandemic has impacted oncological care [32]. Lung cancer patients have increased susceptibility and mortality associated with COVID-19 infection [33,34,35,36]. Early-phase SARS-CoV-2 infections are generally localized to the upper respiratory tract, with progressive infection, in a subset of infected individuals, involving the lower respiratory tract. This leads to excessive cytokine production in the lungs, sustained inflammation and lung injury, which may further deteriorate lung cancer patients’ health. This is evidenced by data showing that the estimated risk of infection from SARS-CoV-2 with severe or fatal complications is approximately 2.3 times higher in lung cancer patients than the general population, or compared to patients with other malignancies. COVID-19 can impact patients with cancer by reducing their functional status, increasing immunosuppression and worsening overall respiratory symptoms, which increases risks of hospitalization and chemotherapy interruptions and overall increases mortality [37,38,39] (Figure 2). It is known that acquired immunity plays a major role in defending against viral infections and cancer [40,41]. Studies have shown decreased overall numbers of natural killer (NK) cells and CD8+ T-cells and increased numbers of hyperactivated CD8+ T-cells in patients with SARS-CoV-2 infection, suggesting that the virus may disrupt the antiviral immunity of the host [42]. This leads to the possibility of immune checkpoint inhibition (ICI) providing protection against the development of severe COVID-19 by preventing lymphocyte exhaustion [43]. 

The potential long-term benefits or harmful effects of ICIs in patients with cancer and COVID-19 infection are not well known and still must be investigated. It is vital to understand the full scope and impact of ICI treatment in COVID-19 patients, to provide better quality of care. Thus, it is the aim of our retrospective chart review to describe the disease course and identify specific risk factors and overall outcomes in a very diverse patient population with an underlying malignancy and COVID-19 infection who received IMT or chemotherapy.

## 2. Methods

### 2.1. Study Design and Population

We performed a retrospective chart review of 121 patients (age > 18 years) with a solid malignancy at the University of Alabama at Birmingham (UAB) between January 2020 and November 2021. Any patients with an established stage 3 or 4 solid malignancy diagnosis, apart from breast, prostate cancer and gastrointestinal stromal tumors (GIST), who tested positive for SARS-CoV-2 through nasopharyngeal swab nucleic acid amplification or serological testing, were included in this study. Asymptomatic individuals who did not require hospitalization were also included. Of the 121 patients in this review, 61 received IMT treatment within 12 months. Treatment response was assessed per iRECIST criteria [44]. Clinical and laboratory data were obtained in full accordance with UAB’s Institutional Review Board (IRB) and the Health Insurance Portability and Accountability Act (HIPAA) Privacy Rule. The declassified data collected were stored on a UAB encrypted and password protected share-drive. UAB encrypted and password protected share-drive Any deviation from the protocol was documented and reported accordingly.

### 2.2. Data Collection

Patient demographics such as age, gender and race were collected. In addition, recorded clinical characteristics included age at COVID-19 diagnosis, smoking history, comorbidities (cardiovascular, pulmonary, hepatic, renal disease, obesity, venous thromboembolism, secondary malignancy or diabetes mellitus), date of COVID-19 diagnosis, history of COVID-19 vaccination, cancer characteristics (stage, type, date of diagnosis, time between COVID-19 diagnosis and last IMT administration, programmed cell death ligand 1 (PD-L1) status (NA = not performed, 0–100 range), immune-related adverse events (irAE), Eastern Cooperative Oncology Group (ECOG) performance status and COVID-19 symptoms (fever defined as measurement greater than 37 °C, cough, dyspnea)).

Laboratory blood work in the study included baseline white blood cells (WBC), WBC during initial COVID-19 diagnosis, lymphocyte count during initial COVID-19 diagnosis, c-reactive protein (CRP), creatinine, estimated glomerular filtration rate (eGFR), aspartate transaminase (AST), alanine transaminase (ALT), lactose dehydrogenase (LDH) and d-dimer. Initial imaging was also documented with the most common finding (presence and location of opacities). Oxygen saturation by pulse oximeter (SpO2) was measured at admission. Admission (COVID-19 floor or intensive care unit (ICU)) and treatment (oxygen therapy, mechanical ventilation, use of vasopressors, continuous renal replacement therapy (CRRT), antiviral therapy, antibiotics and glucocorticoids) of COVID-19 were conducted according to UAB’s institutional protocol and at the treating provider’s discretion. 

Clinical outcomes measured included oxygen requirement, hospitalization rate, ICU admission, use of mechanical ventilation, death of any cause and overall survival (OS). OS or “days from date of pathological diagnosis” was defined as the period from the date of diagnosis until the last office visit. “Days from end of treatment” measured the time from COVID-19 diagnosis until the last office visit.

### 2.3. Statistical Analysis

Frequency and proportion calculations were used to describe patient demographics and overall characteristics. Chi-square tests were utilized to assess differences among the groups for categorical variables, and *t*-tests analyzed the differences between groups for continuous measures (Table 1, Table 2, Table 3, Table 4 and Table 5). 

## 3. Results

### 3.1. Baseline Characteristics

Sample descriptive statistics can be found in Table 1. A total of 121 patients were examined in this study and 61 received IMT treatment within 12 months. Fifty-six percent of patients received pembrolizumab, 15.0% nivolumab, 18.0% atezolizumab, 7.0% durvalumab, 1.0% ipilimumab and 3.0% ipilimumab/nivolumab. The three most common irAEs in decreasing order were endocrine (6 cases; 4 adrenal insufficiency, 1 hypophysitis and 1 hypothyroid), gastrointestinal (5 cases; 5 colitis) and skin (4 cases; 3 rash and 1 psoriasis). The median age at diagnosis for IMT patients was 62 years, 62.0% were male, and 66.0% of the study population were white. For the 60 patients on chemotherapy, the median age at diagnosis was 65 years and 53.0% were male. Seventy-three percent of the study population were white and 62.0% were former smokers (>12 months), 25.0% never smoked and 13.0% were current smokers in the IMT group (Appendix A). Meanwhile, in the chemotherapy cohort, 48.0% were former smokers (>12 months), 32.0% were never smokers and 20.0% were current smokers (Table 2). Of the 61 patients treated with IMT, the most common cancer was lung (33.0%), followed by hepatocellular carcinoma (HCC, 13.0%) and renal cell carcinoma (RCC, 11.0%, Table 1 and Appendix A). Hypertension (49.0%), diabetes mellitus (26.0%), hyperlipidemia (21.0%) and secondary malignancies (18.0%) were the most common comorbidities in decreasing order. Within the 60 patients on chemotherapy, the most common cancers were lung (33.0%), HCC (12.0%) and head and neck cancers (10.0%). The most common comorbidities in the same patients, in decreasing order, were hypertension (53.0%), diabetes mellitus (28.0%), hyperlipidemia (23.0%) and COPD (18.0%). The PD-L1 expression in the IMT group was 15, compared to 11 in the chemotherapy group. Moreover, the ECOG was 1.0, versus 1.1 in the IMT and chemotherapy groups, respectively. Additionally, 56.0% of the IMT cohort received a COVID-19 vaccine (8 people received a vaccine before their diagnosis), compared to 62.0% in the chemotherapy group (9 people received a vaccine before their diagnosis; Table 3 and Appendix A).

### 3.2. Clinical Course

The average duration between COVID-19 diagnosis and ICI was 3.9 months. Approximately 60.0% of the IMT group had symptoms (59.0%), with the most common symptoms being dyspnea (47.0%), cough (33.0%) and fatigue (11.0%). In comparison, three quarters of the patients treated with chemotherapy had symptoms, including dyspnea (54.0%), cough (23.0%) and fever (23.0%, Table 1). The average WBC for IMT patients was 7.4, with a baseline WBC during COVID-19 infection of 7.7 and a lymphocyte count of 1015.0. Patients treated with chemotherapy had a baseline WBC of 7.1, with an average WBC of 7.1 during COVID-19 and a lymphocyte count of 967.0 (Table 4). Other pertinent labs (average) for the IMT patients included creatinine 1.3, AST 44.6, ALT 30.2, LDH 354.0, d-dimer 1135.8 and CRP of 90.5. For patients on chemotherapy, the average labs were as follows: creatinine 1.1, AST 41.5, ALT 31.8, LDH 324.0, d-dimer 1232.0 and CRP of 82.4 (Table 4). 

Over half of the IMT patients required hospitalization (53.0%) and six (11.0%) were admitted to the ICU (Figure 3). Moreover, 14 (23.0%) required oxygen supplementation and no patient was intubated. In comparison, 34 (56.0%) of the patients on chemotherapy required hospitalization and three (5.0%) needed specialized care in the ICU (Figure 3, Appendix A). One fifth of the patients on chemotherapy required oxygen supplementation and two (3%) were mechanically ventilated (Appendix A). 

There were also some differences in COVID-19 treatment between the patients treated with ICI and chemotherapy. In the ICI cohort, no patients received antibody/IL-6 therapy, one (2.0%) person required vasopressor support, one (2.0%) patient was on CRRT, nine (15.0%) received broad antibiotics, nine (15.0%) were on remdesivir and 19 (31.0%) received dexamethasone. For patients on chemotherapy, five (8.0%) patients received antibody/IL-6 therapy, two (3.0%) people required vasopressor support, no patients were on CRRT, 14 (23.0%) patients received broad antibiotics, nine (15.0%) were on remdesivir and 12 (20.0%) received dexamethasone.

### 3.3. Risk Factors

There were several risk factors associated with hospitalization. Patients who identified as white were at a significantly increased risk for overall hospitalization in lower acuity units compared to black patients (62.9% vs. 37.1%, *X^2^* = 13.3, *p* < 0.01; Table 1). Higher admissions were also seen in patients with increased frailty, as evidenced by the higher ECOG (1.1 vs. 1.7, 1.0; *t* = 2.4; *p* = 0.05). Specific cancer subtypes excluding NSCLC, liver, RCC and H&N were associated with increased admission (*X^2^* = 13.5, *p* = 0.03). Several specific laboratory findings were associated with increased admission. Patients admitted to lower acuity units had a mean AST value of 40.9, compared to 99.1 for patients in critical care units (*t* = 2.8; *p* < 0.001). Patients admitted to the ICU were more likely to be unvaccinated compared to those admitted to the lower acuity units (1.4% vs. 15.7%; *X^2^* = 10.2, *p* < 0.001). Oxygen supplementation was also higher in critically ill patients in the ICU (88.9% vs. 27.7, *X^2^* = 13.5; *p* < 0.001) compared to the lower acuity units. Treatment-specific risk factors associated with admission to the ICU compared to a regular floor unit included antibiotics (77.8% vs. 30.0, *X^2^* = 8.0, *p* < 0.001), antivirals (66.7% vs. 17.1, *X^2^* = 11.1, *p* < 0.001) and steroids (77.8% vs. 34.3, *X^2^* = 6.3, *p* < 0.01). Smoking status was not associated with treatment outcomes or hospitalization. 

### 3.4. Survival Outcomes

Survival outcomes differed between the ICI cohort and chemotherapy cohort (Table 4). In the ICI cohort, 15 patients died, but only five deaths were directly related to COVID-19 (Appendix A). In comparison, eight patients in the non-immunotherapy group died, and four deaths were directly attributed to COVID-19 (Appendix A). The OS in the ICI cohort was 780 days, compared to 706.0 days in the chemotherapy cohort (t = −0.6, *p* = 0.6). The average time since COVID-19 diagnosis in non-smokers was 159 daysversus 220.0 days smokers. There was an association between reduced mortality in patients with a lower ECOG (<1) versus those patients that were deceased (1.5; *t* = 3.2; *p* < 0.01). Other factors associated with worse mortality were higher oxygen needs (*X^2^* = 13.5, *p* < 0.001), being unvaccinated for COVID-19 (*X^2^* = 15.2, *p* < 0.001), the use of antiviral therapy (*X^2^* = 5.4, *p* < 0.05) and the use of steroids (*X^2^* = 4.3, *p* < 0.1). No mortality differences were observed according to smoking history, types of comorbidities or cancer origin, including “hot” vs. “cold” tumors (Appendix A).

## 4. Discussion

Patients with cancer who were diagnosed with COVID-19 and were treated with ICIs had increased mortality compared to the same type of patients treated with chemotherapy. However, in the ICI cohort, the COVID-19-associated mortality accounted for one third of the cases, while, in the chemotherapy group, it accounted for one half of the cases. The overall mortality was 19% in our patient cohort, which is much higher than the case fatality rate of 3% in the general community [45]. In several other studies of patients with cancer and COVID-19, the 30-day mortality was reported as an average range of 10–40%, which is similar to our results [46,47,48]. 

A systemic review and meta-analysis of COVID-19 patients with cancer who were treated with chemotherapy found that there was an increased risk of death (OR: 1.9; 95% confidence interval: 1.3–2.7), after adjusting for confounding variables, compared to patients that received targeted therapies, IMT, surgery or radiotherapy [49]. Another meta-analysis of 6042 patients with COVID-19 showed that ICIs received within 30 days before diagnosis did not increase mortality or the severity of disease in affected patients [50]. In our review, there was one study that found a tendency of an increasing risk of mortality and severity of critical disease in COVID-19 patients treated with ICIs [51]. This could be related to the results being unadjusted and confounded by several variables (e.g., age, ECOG, comorbidities, smoking, presence of metastasis) in patients diagnosed with COVID-19. It should be noted that smoking increases the risk for intubation and mortality, but, due to our smaller sample size, this was not seen in our review [52]. 

We found that there were several risk factors associated with increased mortality. The factors that lead to worse disease severity and thus reduced OS were increased oxygen needs, being unvaccinated for COVID-19 and the use of antiviral therapy and steroids. However, one very important risk factor that led to worse mortality was an ECOG >1. It is known that a decline in functional status in patients with malignancies can result in an adverse outcome, but an ECOG of 2 is not a contraindication to treatment [53]. Patients with COVID-19 and an ECOG of 2 might require the postponement of therapy until there is an improvement in functional status, as it could potentially worsen outcomes. 

One very important question that providers face is related to the benefit of postponing or temporarily discontinuing ICI therapy in patients with a solid malignancy. It is known that multiple office visits for ICI infusions raises the possibility of COVID-19 exposure and possible infection [54]. However, treatment delays in advanced malignancies can also be very detrimental, especially when considering the current improvements that have been made in treating COVID-19 infections. Another important consideration of ICI therapy in advanced or metastatic disease is the possibility of irAEs such as ICI pneumonitis, which can be masked by COVID-19 symptoms. In particular, 39–54% of patients with COVID-19 are symptomatic and require hospitalization [55,56]. In our own study, 53% of COVID-19-affected patients treated with ICIs required admission to either the hospital or ICU. The most common symptom was dyspnea, which can also be seen in ICI pneumonitis, which accounts for a third of cancer-related deaths [57]. Nevertheless, this would appear to be less of a possibility in patients with COVID-19 as the standard-of-care treatment includes steroids, which are also used to treat ICI pneumonitis [58]. 

This retrospective chart review was able to provide valuable insights, but it also had several limitations. One specific constraint was related to the lack of a comparator group for outcomes (hospitalizations and mortality) in patients with an underlying malignancy who did not have COVID-19. Secondly, as with other retrospective chart reviews, selection bias can occur and it can be difficult to distinguish symptoms from COVID-19 and IMT pneumonitis. Detailed exploration of follow-up questions can be difficult to achieve due to limited information that was not obtained during the original chart abstraction. Additionally, with rarer malignancies, the full impact of COVID-19 will be more challenging to identify unless multicenter, long-term studies are conducted so that subgroup analysis can be performed in a more meaningful manner, with a larger pooled cohort (e.g., antibody/IL-6 therapy in thyroid cancer). Our study also had several strengths. One very important aspect of this retrospective chart review was the diverse patient population, with at least 25% of study cohort identifying as black, and there was a large number of ICI-treated patients for a single institutional study. We also examined several important variables, such as smoking history, COVID-19 vaccination status and irAEs. 

## 5. Conclusions/Future Perspectives

ICI mortality was higher compared to patients receiving chemotherapy post-COVID-19. Risk factors for hospitalization and increased disease severity/mortality for COVID-19-affected patients have been identified to aid in future risk stratification. There are still many unknown factors regarding COVID-19 survivors who are treated for an underlying malignancy. “Long COVID” refers to symptoms of COVID-19 that can last for months after the initial diagnosis and can have multi-organ effects. The specific triggers and management in patients with cancer are unknown and should be studied prospectively in larger cohorts.

In our opinion, ICI or chemotherapy cessation or delay is unwarranted if there is a risk–benefit assessment with the patient and there is an understanding that disease progression can be more detrimental. However, further investigation still needs to be undertaken to understand whether the PD-L1 pathway with the subsequent inflammatory cascade post-COVID-19 can impact overall survival and whether the timing between ICI and COVID-19 infection affects long-term symptoms/outcomes. 

## Figures and Tables

**Figure 1 cancers-14-05954-f001:**
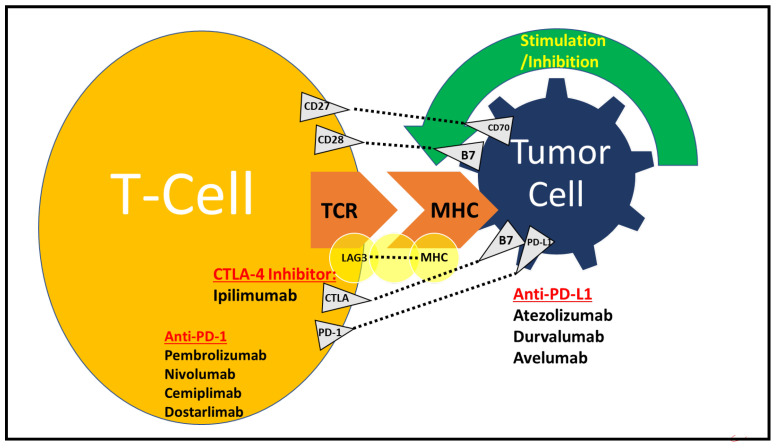
**Mechanism of action of PD-1/PD-L1 and CTLA4 inhibitors.** Antigen presentation occurs on the surfaces of antigen-presenting cells (APCs) via T-cell receptors (TCR) and major histocompatibility complex (MHC). There are stimulatory and inhibitory signals, such as cytotoxic T-lymphocyte-associated protein 4 (CTLA-4) and programmed cell death-1 (PD-1). Some of the upregulators of T-cells and their cognate ligands are CD27-CD70 and CD28-B7. Inhibitory T-cell receptors and their ligands include LAG3-MHC, CTLA4-B7 and PD1-PDL1. CTLA-4 reduces signaling via its co-stimulatory receptor, CD28, by binding to CD80 and CD86 on APC. CTLA-4 sends an inhibitory signal to T-cells. A CTLA-4 inhibitor (ipilimumab) stops autoreactive T-cells. PD-1 binds to programmed death-ligand 1 (PD-L1), which results in cancer evasion from the immune system. Blockade of PD-1 (via pembrolizumab, nivolumab, cemiplimab or dostarlimab) or PD-L1 (via atezolizumab, durvalumab or avelumab) increases anti-tumor immune activity. The image used has been modified from a prior submission [11].

**Figure 2 cancers-14-05954-f002:**
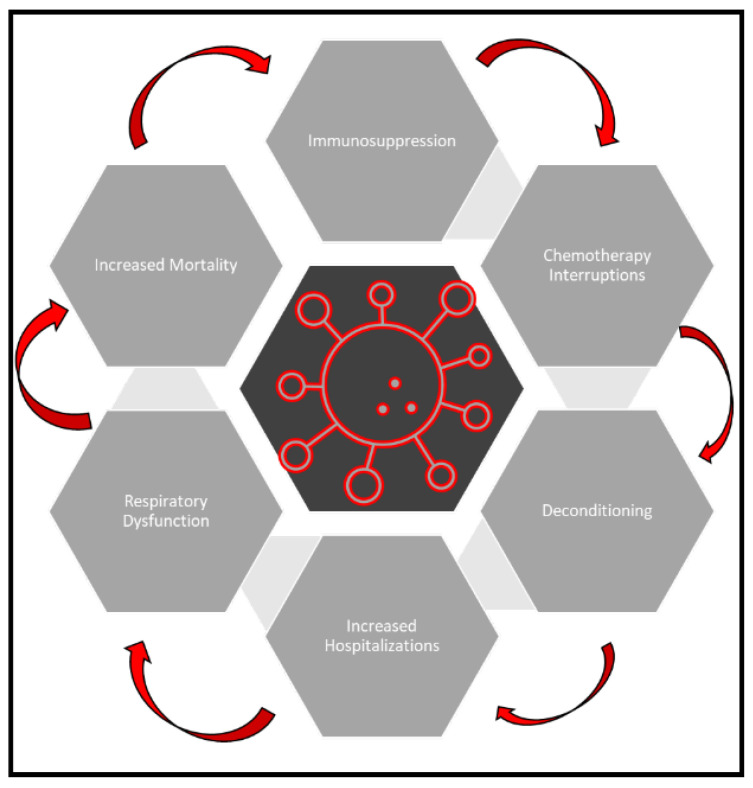
**Clinical outcomes of COVID-19 in cancer patients.** COVID-19 can cause functional decline, respiratory distress and immunosuppression, which increases risks of hospitalization, chemotherapy interruptions and overall mortality.

**Figure 3 cancers-14-05954-f003:**
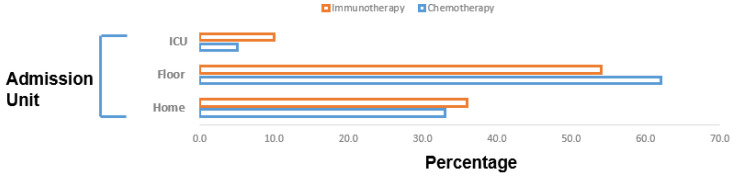
Admission unit of COVID-19-affected patients (*n* = 121). In total, 32 (53.0%) of the immunotherapy patients required hospitalization and 6 (11.0%) were admitted to the ICU. For the patients on chemotherapy, 34 (56.0%) required hospitalization and 3 (5.0%) needed specialized care in the ICU.

**Table 1 cancers-14-05954-t001:** Characteristics of the sample, COVID-19 infection, all cancers (*n* = 121).

Patient Characteristics	Treatment			
	Sample	Chemotherapy (*n* = 60)	Immunotherapy (*n* = 61)	*t/Χ^2^*
**Sociodemographics**				
Age (M/SD)	63.7 (11.6)	65.1 (12.9)	62.3 (10.0)	1.4
Sex				
Female	51.0 (42.2%)	28.0 (46.7%)	23.0 (37.7%)	
Male	70.0 (57.9%)	32.0 (53.3%)	38.0 (62.3%)	1.0
Race/Ethnicity				
Black	35.0(28.93%)	16 (26.67%)	19.0 (31.2%)	
White	84.0 (69.4%)	44.0 (73.3%)	40.0 (65.6%)	
Asian/Latinx/Other	2.0 (0.2%)	0.0 (0.00%)	2.0 (3.3%)	2.4
**Clinical Factors**				
Comorbidities				
0	4.0 (3.3%)	1.0 (1.7%)	3.0 (4.9%)	
1–2	15.0 (12.4%)	7.0 (11.4%)	8.0 (13.1%)	
3–5	64.0 (52.9%)	30.0 (50.0%)	34.0 (55.7%)	
>5	38.0 (31.4%)	22.0 (36.4%)	16.0 (26.2%)	2.3
ECOG				
0–1	96.0 (79.3%)	48.0 (80.0%)	48.0 (78.7%)	
2–4	25.0 (20.7%)	12.0 (20.0%)	13.0 (21.3%)	0.03
Cancer Type				
Lung	40.0 (33.1%)	20.0 (33.3%)	20.0 (33.3%)	
Liver	15.0 (12.4%)	7.0 (11.7%)	8.0 (13.1%)	
Renal	10.0 (8.3%)	3.0 (5.0%)	7.0 (11.4%)	
Head and Neck	11.0 (9.1%)	6.0 (10.0%)	5.0 (8.2%)	
Other	45.0 (37.2%)	24.0 (40.0%)	21.0 (34.4%)	2.0
Symptoms at COVID-19 Diagnosis				
Fever	19.0 (21.6%)	12.0 (23.1%)	7.0 (19.4%)	1.1
Cough	24.0 (27.3%)	12.0 (23.1%)	12.0 (33.3%)	0.002
Dyspnea	45.0 (51.1%)	28.0 (53.9%)	17.0 (47.2%)	4.6 *
PD-L1 Expression (M/SD)	13.1 (26.6)	11.4 (27.2)	14.9 (26.6)	0.7
Immunotherapy				
PD-1/PD-L1		--	58.0 (95.1%)	
CTLA-4		--	1.0 (1.6%)	
Combined		--	2.0 (3.3%)	
Admission				
Home	42.0 (34.7%)	20.0 (33.3%)	22.0 (36.1%)	
Floor	70.0 (57.9%)	37.0 (61.7%)	33.0 (54.1%)	
Intensive Care	9.0 (7.4%)	3.0 (5.0%)	6.0 (9.8%)	1.3
Oxygen Use	28.0 (23.1%)	14.0 (23.3%)	14.0 (23.0%)	0.003
Mechanical Ventilation	2.0 (3.3%)	0.0 (0.0%)	2.0 (1.7%)	2.1
Treatment				
Steroids	31.0 (25.62%)	12.0 (20.0%)	19.0 (31.2%)	2.0
Antiviral	18.0 (14.9%)	9.0 (15.0%)	9.0 (14.8%)	0.001
Antibiotics	28.0 (23.1%)	14.0 (23.3%)	14.0 (23.0%)	0.003
Mortality	23.0 (19.0%)	8.0 (13.3%)	15.0 (24.6%)	2.5

** p* < 0.05.

**Table 2 cancers-14-05954-t002:** Associations between clinical factors and smoking status, COVID-19 infection (*n* = 121).

Patient Characteristics	Not a Smoker	Current Smoker	Former Smoker	*X^2^/t/F*
**Clinical Factors**				
Admission				
Floor	15.0/44.1%	12.0/60.0%	43.0/64.2%	
Home	18.0/52.9%	7.0/35.0%	17.0/25.4%	
MICU	1.0/2.9%	1.0/5.0%	7/10.45%	8.4
Days from Cancer Dx to COVID-19	629.6/652.3	342.2/454.0	585.1/821.5	1.1
Days from end of COVID-19 treatment	159.2/128.3	196.7/126.1	260.7/204.7	8.6 *
Oxygen Use	7.0/20.6%	4.0/20.0%	17.0/25.3%	5.0
Treatment				
Steroids	8.0/25.5%	3.0/15.0%	20.0/29.9%	1.9
Antiviral	8.0/14.7%	2.0/10.0%	11.0/16.4%	0.5
Antibiotics	5.0/14.7%	4/20.00%	19/28.36%	2.5
Death	6.0/17.7%	3.0/15.0%	14.0/20.9%	0.4

** p* < 0.05.

**Table 3 cancers-14-05954-t003:** Association between clinical factors and COVID-19 vaccination status, COVID-19 infection (*n* = 121).

Patient Characteristics	COVID-19 Vaccination	No COVID-19 Vaccination	*X^2^* or *t*
**Clinical Factors**			
Admission			
Floor	40.0/57.1%	30.0/58.8%	
Home	29.0/41.4%	13.0/25.5%	
MICU	1.0/1.4%	8.0/15.7%	10.2 **
Days from Cancer Dx to COVID-19	510.3/696.4	622.1/722.3	0.8
Days from end of COVID-19 treatment	230.0/144.6	126.7/127.5	4.1 ***
Oxygen Use	19.0/37.2%	9.0/12.9%	11.3 *
Treatment			
Steroids	15.0/21.4%	16.0/31.4%	1.5
Antiviral	6.0/8.6%	12.0/23.5%	5.2 *
Antibiotics	12.0/17.1%	16.0/31.4%	3.4
Death	5.0/7.1%	18.0/35.3%	15.2 ***

** p* < 0.05, ** *p* < 0.01, *** *p* < 0.001.

**Table 4 cancers-14-05954-t004:** Risk factors of admission, ICU and mortality, COVID-19 infection (*n* = 121).

Patient Characteristics	Admission	ICU	Death
Immunotherapy—61	28.0 (45.9%)	6.0 (9.8%)	15.0 (24.6%) ^
Chemotherapy—60	23.0 (38.3%)	3.0 (5.0%)	8.0 (13.3%) ^
**Sociodemographics**			
Age (M/SD)	64.6 (8.7)	63.0 (11.4)	63.0 (11.4)
Sex			
Female	44.0 (62.9%)	7.0 (77.8%)	8.0 (34.8%)
Male	26.0 (51.0%)	2.0 (22.2%)	15.0 (65.2%)
Race/Ethnicity			
Black	26.0 (37.1%)	2.0 (22.2%)	6.0 (26.1%)
White	44.0 (62.9%)	6.0 (22.2%)	16.0 (69.6%)
Asian/Latinx/Other	0.0 (0.0%) *	1.0 (11.11%)	1.0 (4.35%)
**Clinical Factors**			
Comorbidities			
0	1.0 (1.4%)	0.0 (0.0%)	0.0 (0.0%)
1–2	10.0 (14.3%)	2.0 (22.2%)	4.0 (17.4%)
3–5	38.0 (54.3.0%)	5.0 (55.6%)	10.0 (43.5%)
>5	21.0 (30.0%)	2.0 (22.2%)	9.0 (39.1%)
ECOG	1.0 (0.8)	1.7 (0.7)	1.5 (0.8)
Symptoms at COVID-19 Diagnosis			
Fever	11.0 (21.6%) **	7.0 (77.8%) **	33.0 (33.4%)
Cough	5.0 (19.2%)	1.0 (11.1%)	4.0 (17.4%)
Dyspnea	8.0 (15.7%)	4.0 (44.4%) *	4.0 (17.4%)
PD-L1 Expression (M/SD)	12.8 (27.0)	19.1 (33.7)	13.0 (23.6)
Immunotherapy			
PD-1/PD-L1	31.0 (93.9%)	6.0 (100.0%)	14.0 (93.3%)
CTLA-4	0.0 (0.0%)	--	1 (6.7%)
Combined	2.0 (6.1%)	--	0 (0.00%)
AST	40.9 (48.3)	99.1 (94.6) **	50.0 (65.4)
LDH	341.1 (317.9)	564.3 (436.3)	381.1 (327.0)
Oxygen Use	19.0 (27.7%)	8.0 (88.9%) ***	12.0 (52.2%) ***
Treatment			
Steroids	27.0 (34.3%)	7.0 (77.8%) *	10.0 (43.5%) *
Antiviral	12.0 (17.1%)	6.0 (66.7%) ***	7.0 (30.4%) *
Antibiotics	21.0 (30.0%) *	7.0 (77.8%) ***	8.0 (34.8%)

** p* < 0.05, ** *p* < 0.01, *** *p* < 0.001, ^ chi-square = 2.5, *p* = 0.2.

**Table 5 cancers-14-05954-t005:** Risk factors of admission, ICU and mortality, COVID-19 infection (*n* = 121).

Patient Characteristic	Outcomes		
	Admission	ICU	Death
Cancer			
Lung	27/38.57%	4/44.44%	7/30.43%
Liver	8/11.43%	1/11.11%	0/0.00%
Renal	9/12.86%	0/0.00%	3/13.04%
Head and Neck	2/2.86%	1/11.11%	1/4.35%
Other	24/34.28% *	3/33.34%	12/52.18%

** p* < 0.05.

## Data Availability

The de-identified patient data used in this study can be requested from Dr. Jessy Deshane at the University of Alabama at Birmingham.

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
