# Peer review of "Clinical Outcomes in COVID-19 Patients Treated with Immunotherapy"

_cancers, 2022, doi:10.3390/cancers14235954_

Round 1

Reviewer 1 Report

The manuscript entitled “Clinical Outcomes in COVID Patients Treated with Immuno-therapy” by Hatic H. and colleagues evaluated, in a retrospective study, whether patients with cancer who get diagnosed with COVID-19 and treated with ICIs present a mortality rate different than patients treated with chemotherapy. They found that cancer patients treated with ICI had a higher percentage of fatal condition compared the cancer patients treated with chemotherapy. However, cancer patients, who received a COVID-19 vaccine, showed lower mortality compared to unvaccinated patients. Nonetheless, patients with lower ECOG had a lower risk of mortality compared to patients with higher functional status.

This is a small retrospective study, in which the authors confirmed data already established in the scientific field, and that COVID-19 infection in ICI treated patients expose the patients to a higher risk of mortality.   

 I have some comments for the authors.

1) ICI has a different impact and efficacy depending on the type of tumor (e.g. cold and hot tumors). What is the outcome of patients with different type of cancer, treated with ICI and being infected with Sars-Cov-2?

2) How is the ICI efficacy evaluated?

3) The authors should comment on co-infection events and thrombotic episodes that might have occurred in the ICI and chemotherapy treated groups.

4) Can the authors address/state the limitations of the study?

5) Can comorbidities be considered in a mortality multivariate analysis?

Author Response

Reviewer#1:

  1. ICI has a different impact and efficacy depending on the type of tumor (e.g. cold and hot tumors). What is the outcome of patients with different type of cancer, treated with ICI and being infected with Sars-Cov-2?

Thank you for this suggestion. We compared nonimmunogenic “cold” tumors (ovarian, pancreatic) to malignancies that are known to be infiltrated by T-cells or “hot” tumors (melanoma, bladder, kidney, head & neck and non-small cell lung cancer). Afterwards we examined the impact (hospital admission, ICU admission, and death) to the tumor type (hot vs cold). None of the endpoints for outcomes were significant.

  1. How is the ICI efficacy evaluated?

We have updated the “Methods” section on page 13 to reflect how the evaluation of ICI is conducted. The iRECIST criteria was employed to assess response to ICI.

  1. The authors should comment on co-infection events and thrombotic episodes that might have occurred in the ICI and chemotherapy treated groups.

18 total patients (9 on IMT and 9 on chemotherapy) required antiviral treatment and 28 patients were initiated on antibiotics (14 on IMT and 14 on chemotherapy). In addition, there were 5 patients in the ICI group and 7 patients in the chemotherapy cohort with VTE. However, due to the limitation of a retrospective chart review analysis, we were not able to obtain information on specific type of co-infections and acuity of thrombotic episodes.   

  1. Can the authors address/state the limitations of the study?

The one major limitation of the retrospective chart review was a lack of a comparator group for outcomes in patients uninfected by COVID-19. Detailed exploration of follow-up questions can be difficult to achieve due to lacking information. Also, selection bias can be hard to exclude. Lastly, it can be difficult to distinguish symptoms from COVID-19 and IMT pneumonitis and full impact of the rare malignancies can be hard to identify. Please see pages 22 of the “Discussion” section.

  1. Can comorbidities be considered in a mortality multivariate analysis?

We examined the major comorbidities (cardiac, renal, hepatic, pulmonary, venous thromboembolisms, diabetes, obesity, and secondary malignancies) via a mortality multivariate analysis and did not find any significant outcomes.  

Reviewer 2 Report

in the present report the Authors have retrospectively evaluated the impact of two different therapeutic approaches on clinical outcome in patients with Covid-19 infection. The conclude that therapy with immune check point inhibitors was associated with a worst outcome in these patients as compared with chemotherapy. 

The report faces with an interesting topic about a selected patient population that might be considered as fragile and that is not completely investigated in the context of Covid-19 infection. Although this very interesting background, the report in the present form has several flaws that should be addressed before being accepted for publication.

Comments and concerns.

1. In the simple summary, the Authors claim that one of the aim of the study was to evaluate the hypothetical association between smoking status, Covid 19 vaccination and clinical factors, but these observations are missed in the manuscript probably because these associations have not been investigated and shown. 

2. A short paragraph and a figure about the mechanisms of action of immune check point inhibitors should be added.

3. Similarly, a description and a figure about how Covid-19 infection might cause the worse clinical outcome should be added.

4. Which kind of vaccine was administered? Moreover, were there differences in vaccinated patients accordingly to different vaccines?

5. Patients included in the study had different cancer types, have the authors observed differences in clinical outcome accordingly to different cancer types?  

Author Response

  1. In the simple summary, the Authors claim that one of the aim of the study was to evaluate the hypothetical association between smoking status, Covid 19 vaccination and clinical factors, but these observations are missed in the manuscript probably because these associations have not been investigated and shown. 

We have revised the simple summary so that the overall aim of this retrospective review reflects our investigation. Associations between smoking status, COVID-19 vaccination and clinical factors have been highlighted in the “Results” section on pages 14-21. We have also added data regarding smoking status, COVID-19 vaccination, and other clinical factors to the “Supplement” for a fuller picture.

  1. A short paragraph and a figure about the mechanisms of action of immune check point inhibitors should be added.

We appreciate this feedback!! 2 paragraphs and a figure about the mechanism of action of ICI was added in the “Introduction” section on page 10-11.

  1. Similarly, a description and a figure about how Covid-19 infection might cause the worse clinical outcome should be added.

Thank you for your feedback. We have added a description of the deleterious effects of COVID-19 infection on clinical outcomes. Please see the “Introduction” section of our paper on page 10-11.

  1. Which kind of vaccine was administered? Moreover, were there differences in vaccinated patients accordingly to different vaccines?

The vaccines included: Pfizer-BioNTech, Moderna and Johnson & Johnson’s Janssen. Our institution primarily offers the Pfizer-BioNTech COVID-19 vaccine. Due to the nature of a retrospective chart analysis, we were unable to assess for differences in patients per specific vaccine type as these data was not available in our patient cohort. Vaccination status was obtained, but not the type of vaccine.   

  1. Patients included in the study had different cancer types, have the authors observed differences in clinical outcome accordingly to different cancer types?

We ran a series of logistic regressions predicting hospital admission, ICU admission, and death to test for the association of type of tumor, treatment, and COVID-19 infection, and the interactive effects.  None of the main effects were significantly associated with the outcomes and thus, the interactive effects were also not significant.

Reviewer 3 Report

Two points will need to be addressed in my opinion prior to acceptance:

1. Please place the statistical significance (or lack thereof) comparing the deaths among the ICI (34.59%) and chemotherapy (13.33%) cohorts in Table 4. 

2. For clearer reading would round the numbers to the nearest tenth (not one hundredth). 

3. For clarity it would be recommended to place "61"  just to the right of "Immunotherpy"and "60" just to the right of Chemotherapy under Patient Characteristics in Table 4.

4. Recommend adding the COVID-associated mortality in each cohort to Table 4. Also, the second sentence of the discussion is somewhat unclear stating that the Covid associated mortality in the ICI group accounted for one-in-3 of the cases - would change this to "one third" as the COVID associated mortality in the chemotherapy cohort was written as "one-half".

Author Response

  1. Please place the statistical significance (or lack thereof) comparing the deaths among the ICI (34.59%) and chemotherapy (13.33%) cohorts in Table 4. 

Statistical significance has been incorporated to the mortality analysis of the ICI vs chemotherapy cohort in Table 4 on pages 18-19. It reads 54.59% instead of 34.59%. Chi-square = 2.49, p=0.16.

  1. For clearer reading would round the numbers to the nearest tenth (not one hundredth). 

We appreciate this suggestion! All the numbers have been rounded to the nearest tenth as per your recommendation. Please see Tables 1-4 (pages 15-20).

  1. For clarity it would be recommended to place "61"  just to the right of "Immunotherpy"and "60" just to the right of Chemotherapy under Patient Characteristics in Table 4.

We have placed the numbers 61 and 60 to the right of “Immunotherapy” and “Chemotherapy” respectively in Table 4 on pages 18-19.

  1. Recommend adding the COVID-associated mortality in each cohort to Table 4. Also, the second sentence of the discussion is somewhat unclear stating that the Covid associated mortality in the ICI group accounted for one-in-3 of the cases - would change this to "one third" as the COVID associated mortality in the chemotherapy cohort was written as "one-half".

Thank you for the recommendation. 5 out of the 15 patients in the ICI and 4 out of 8 patients in the chemotherapy group had specific COVID-associated mortality. In addition, we have rephrased the sentence that described the COVID associated mortality on page 18 of the “Discussion “section.

Round 2

Reviewer 1 Report

The authors provided sufficient explanation to my questions.